# Model Predictive Control Strategy for the Degradation of Pharmaceutically Active Compounds by UV/H$_2$O$_2$ Oxidation Process

**Juwon Lee** [1,2]**, Sook-Hyun Nam** [2]**, Jae-Wuk Koo** [2]**, Eunju Kim** [2] **and Tae-Mun Hwang** [1,2,]*

1    Civil and Environmental Engineering, Korea University of Science & Technology, 217 Gajung-ro, Yuseong-gu, Daejeon 305-333, Korea; leejw0410@kict.re.kr
2    Korea Institute of Civil Engineering and Building Technology, 283 Goyangdar-Ro, Ilsan-Gu, Goyang-si 411-712, Korea; fpnsh@kict.re.kr (S.-H.N.); koojaewuk@kict.re.kr (J.-W.K.); kej@kict.re.kr (E.K.)
*    Correspondence: taemun@kict.re.kr; Tel.: +82-031-910-0741

**Abstract:** Hydroxyl radical (•OH) scavenging demand can be an indicator that represents the water quality characteristics of raw water. It is one of the key parameters predicting UV/H$_2$O$_2$ system performance and affects the operating parameters. Based on the •OH scavenging demand, we developed a model predictive control strategy to meet the target compound removal efficiency and energy consumption simultaneously. Selected pharmaceutically active compounds (PhACs) were classified into three groups depending on the UV direct photolysis and susceptibility to •OH. Group 1 for photo-susceptible PhACs (acetaminophen, amoxicillin, diclofenac, iopromide, ketoprofen, and sulfamethoxazole); group 2 for PhACs susceptible to both direct photolysis and •OH oxidation (bisphenol A, carbamazepine, ibuprofen, naproxen, ciprofloxacin, and tetracycline); and group 3 for photo-resistant PhACs (atenolol, atrazine, caffeine, and nitrobenzene). The results of modeling to achieve 90% removal of PhACs at N and B plants were as follows. For group 2, the optimized operating parameter ranges were as follow (N plant: UV 510–702 mJ cm$^{-2}$, H$_2$O$_2$ 2.96–3.80 mg L$^{-1}$, EED 1088–1302 kWh m$^{-3}$; B plant: UV dose 1179–1397 mJ cm$^{-2}$, H$_2$O$_2$ dose 3.56–7.44 mg L$^{-1}$, EED 1712–2085 kWh m$^{-3}$). It was confirmed that the optimal operating conditions and EED values changed according to the •OH scavenging demand.

**Keywords:** ultraviolet advanced oxidation process; pharmaceuticals; hydroxyl radical; scavenging demand; model predictive control

## 1. Introduction

In recent years, studying the pollution levels in drinking water, especially the presence of pharmaceutical compounds in the environment, has increasingly garnered interest [1]. Presently, there are many pharmaceutically active compounds (PhACs) in the aquatic environment that threaten human and animal health [2]. Over the past decades, the problems of PhACs in the water environment have been widely discussed in many countries. Many studies have found the presence of PhACs in wastewater, which has been identified as a significant source of medicinal substances in drinking water and owing to the intentional and continuous use of water by humans, large quantities of PhACs have been introduced into the environment [3,4]. These substances are known to enter and remain in rivers or lakes, have long periods of biological activity, and act as potential hazards in ecosystems [5]. There are 38 kinds of medicinal substances and five kinds of medicinal metabolites that have been detected in the water system of the Han River, Korea, and 41 types of medicinal substances and five kinds of medicinal metabolites were also detected in the national water supply management system [6]. The removal of PhACs may vary depending on the facilities and their physical and chemical properties. According to recent studies, most residual

pharmaceuticals can be removed efficiently through ozonation, activated carbon adsorption, membrane filtration (e.g., reverse osmosis and nanofiltration), catalytic ozonation, and Fenton oxidation [5,7–11].

The advanced oxidation process (AOP) is a water treatment method that maximizes OH radicals with a high oxidation power using ultraviolet (UV) and $H_2O_2$, which is considered as an alternative process to using ozone [12–15]. It is very effective in the oxidation and mineralization of most organic pollutants [16] and has been used worldwide for groundwater and drinking water remediation. Unlike the ozone-activated carbon process, it has fewer required sites, convenient operation automation, and no impact on the corrosion of facilities caused by auxiliary facilities for ozone generation and residual ozone. The $UV/H_2O_2$ process was introduced to two water drinking water facilities in Siheung and Goyang, South Korea. The introduction of $UV/H_2O_2$ is expected to advance oxidation technology, which will continue to improve.

The hydroxyl radical ($\bullet OH$) is a significant oxidant species in the AOP process with strong oxidizing potential and non-selectivity [17]. $\bullet OH$ is determined by the presence and concentration of scavenging demands. $\bullet OH$ scavenging demand represents the $\bullet OH$ scavenging rate of background material in the water matrix and is known as one of the crucial parameters to predict the $UV/H_2O_2$ process [14,18]. $\bullet OH$ scavenging demand can be an indicator that can represent the water quality characteristics of the target raw water; therefore, continuous monitoring is required when the deviation is significant.

In this study, a continuous measurable device for $\bullet OH$ scavenging demand was used to monitor the $\bullet OH$ scavenging characteristics of water from two purification plants. Target PhACs were tested to compare the removal rates under different $\bullet OH$ scavenging demands. A nonlinear model was established that reflects removing the target PhACs and the energy consumption considering the $\bullet OH$ scavenging demand. Considering the photo-decomposition of each PhAC, the model was divided into three influential groups according to UV direct photo-decomposition vulnerability. Optimized UV irradiation and $H_2O_2$ injection amounts for each group were derived considering the electric energy demand (EED, unit: $kWh\ m^{-3}$) value for removing target PhACs through the generalized reduced gradient (GRG) optimization algorithm matrix.

## 2. Materials and Methods

### 2.1. Chemicals

Sixteen PhACs (acetaminophen, amoxicillin, atenolol, atrazine, bisphenol A, caffeine, carbamazepine, ciprofloxacin, diclofenac, ibuprofen, iopromide, ketoprofen, naproxen, nitrobenzene, sulfamethoxazole, and tetracycline) were selected as target compounds to study the removal characteristics of PhACs by direct UV photolysis and $\bullet OH$ oxidation during $UV/H_2O_2$ processes. These substances are pharmaceuticals that exist in rivers [19], and have recently been detected in domestic water systems, and detection cases have been reported [20–22]. The PhACs are classified [2,13,23,24] in Table 1, where the characteristics of each substance [25–27] are summarized. Atenolol, caffeine, carbamazepine, and sulfamethoxazole were used in the lab-scale $UV/H_2O_2$ experiments. PhACs used in the experiments were purchased from CHIRON (1 mL, 1000 μg $mL^{-1}$ in methanol, Trondheim, Norway).

**Table 1.** Photochemical reaction constants of PhACs for direct UV photolysis and •OH oxidation.

| Drug | $\varepsilon$254 nm ($M^{-1}$ $cm^{-1}$) | $\Phi$254 nm (mol $ein^{-1}$) | $k_{•OH, M}$ ($M^{-1}s^{-1}$) | Classification | Chemical Structure |
|---|---|---|---|---|---|
| Acetaminophen | 8095 | 1.8000 | $1.70 \times 10^9$ | Analgesic | |
| Amoxicillin | 1200 | 0.3720 | $5.43 \times 10^9$ | Antibiotic | |
| Atenolol | 300 | 0.0890 | $7.10 \times 10^9$ | Antihypertension | |
| Atrazine | 3400 | 0.0477 | $2.30 \times 10^9$ | Herbicide | |
| Bisphenol A | 750 | 0.0066 | $8.00 \times 10^9$ | Xenoestrogen, hormone-like properties | |
| Caffeine | 3920 | 0.0018 | $6.40 \times 10^9$ | Stimulants | |
| Carbamazepine | 6070 | 0.0006 | $8.02 \times 10^9$ | Anticonvulsant | |
| Ciprofloxacin | 17,200 | 0.0118 | $5.94 \times 10^9$ | Antibiotic | |
| Diclofenac | 4770 | 0.2920 | $8.38 \times 10^9$ | Analgesic | |
| Ibuprofen | 256 | 0.1920 | $7.40 \times 10^9$ | Analgesic | |
| Iopromide | 21,000 | 0.0390 | $3.30 \times 10^9$ | X-ray contrast agent | |
| Ketoprofen | 15,300 | 0.2980 | $6.89 \times 10^9$ | Antibiotic | |
| Naproxen | 4000 | 0.0278 | $8.61 \times 10^9$ | Analgesic | |
| Nitrobenzene | 5560 | 0.0070 | $3.40 \times 10^9$ | Pharmaceutical | |

**Table 1.** *Cont.*

| Drug | ε254 nm (M⁻¹ cm⁻¹) | Φ254 nm (mol ein⁻¹) | k•OH, M (M⁻¹s⁻¹) | Classification | Chemical Structure |
|---|---|---|---|---|---|
| Sulfamethoxazole | 13,200 | 0.0379 | $5.50 \times 10^9$ | Antibiotic | |
| Tetracycline | 8820 | 0.0038 | $7.70 \times 10^9$ | Antibiotic | |

### 2.2. •OH Scavenging Demand Measurement

The •OH scavenging demand was determined experimentally using a spectrophotometric method with rhodamine B (RhB) as a probe compound. A detailed method for the •OH scavenging demand analysis was described in a previous study [28]. Kwon and Hwang have proposed a spectrophotometric method based on the $R_{OH,UV}$ concept that uses Rhodamine B (RhB) to measure the OH radical scavenging demand. The method is presented in the previously published literature [12,18]. The •OH scavenging demand can be calculated by the following equation (Equation (1)) using measured $R_{OH,UV}$ values [29].

$$\sum k_{s,OH}[S]_i = k_{H_2O_2,OH} \cdot \frac{m}{b} - k_{OH,RhB}^{app}[RhB] \tag{1}$$

where $\sum k_{s,OH}[S]_i$ is the scavenging factor (s⁻¹), $k_{H2O2,OH}$ is the second-order rate constant between •OH and $H_2O_2$, $m$ and $b$ are the factors obtained by the $R_{OH,UV}$ values, and [RhB] is the initial concentration of RhB.

This study monitored raw water using a portable •OH scavenging demand analyzer (Figure 1). For the detector, a VIS-NIR Tungsten Halogen Light (360–2000 nm, Ocean Optics, Rochester, NY, USA) was used as a light source to detect changes in the reactor, and a small UV/VIS spectrometer (200–850 nm, Flame, Ocean Optics, Rochester, NY, USA) was used. The incident irradiance measurements were obtained by placing a calibrated radiometer (UVX Radiometer, UVP Co., East Lyme, CT, USA) at the height of the water level in the Petri dish to obtain the average incident irradiance across the solution surface using the Petri dish factor (PF) and reflection factor (RF) [30]. Changes in RhB were continuously measured by connecting fiber (Single Patch Cord, Ocean Optics, USA) and sensors (UV/VIS Collaborating Lens, 200–2000 nm, Ocean Optics, Rochester, NY, USA). The data measured were analyzed using the OceanView spectroscopy software with a graphical user interface (Ocean Optics, Rochester, NY, USA).

### 2.3. Analytical Methods

The target PhACs were analyzed using high-performance liquid chromatography (HPLC) (1290, Agilent, Santa Clara, CA, USA) and MS/MS (6490, Agilent, Santa Clara, CA, USA). C18 (ZORBAX Eclipse Plus, Agilent, Santa Clara, California, USA) with 2.1 × 100 mm, 3.5 μm particle size was used as an analytical column. The analytical conditions for each substance are shown in Table 2. Total organic carbon (TOC) was analyzed with a TOC analyzer (TOC-VCPH/CPN, Shimadzu, Kyoto, Japan). The UV absorbance at 254 nm and the color of the samples were measured by spectrophotometer (DR 5000, HACH, Loveland, CO, USA). Alkalinity was measured at pH 4.5, adjusted by 0.02 N $H_2SO_4$. pH and TDS were measured using a benchtop meter (ORION 3 STAR, Thermo, Waltham, Massachusetts, USA). Total nitrogen (T-N) was analyzed by a multi-parameter photometer (SYNCA 3ch, BLTech, Aichi, Nagoya, Japan). Turbidity was measured with a turbidimeter (2100N, HACH, Loveland, CO, USA).

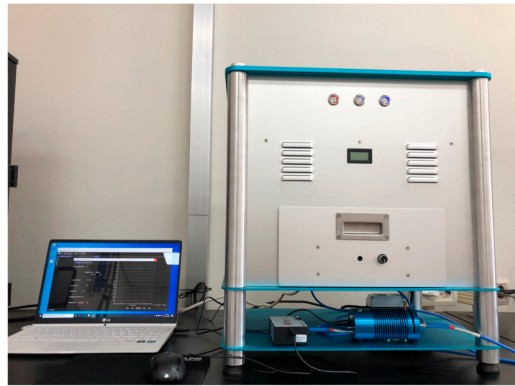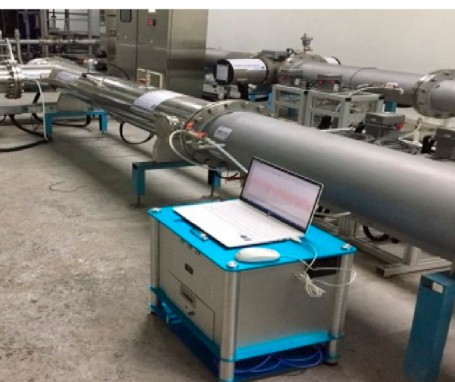

**Figure 1.** Portable •OH scavenging demand analyzer at $UV/H_2O_2$ pilot plant.

**Table 2.** Precursor ion, product ion, and collision energy for the determination of pharmaceuticals.

| Compounds | Precursor Ion (m/z) | Product Ion (m/z) | Collision Energy (eV) |
|---|---|---|---|
| Atenolol | 267.17 | 145.1 [a], 190.1 [b] | 27 [a], 15 [b] |
| Caffeine | 195.09 | 138.1 [a], 110 [b] | 19 [a], 36 [b] |
| Carbamazepine | 237.10 | 194.1 [a], 193.1 [b] | 19 [a], 36 [b] |
| Sulfamethoxazole | 254.06 | 92 [a], 108 [b] | 26 [a], 23 [b] |

[a] Quantitation; [b] confirm ion.

## 3. Results and Discussion

### 3.1. Measurement of Water Quality and OH Radical Scavenging Demand

As the target raw water, sand filtration water from two water purification plants (N and B) with different regions was used for each experiment. The N water purification plant uses river water as its source of water and has a low total organic carbon (TOC) concentration, while the B water purification plant uses a lake as its source and has a relatively high TOC concentration. The TOC concentration of B-filtered water was 6.96 mg L$^{-1}$, which was approximately 3.7 times higher than that of N filtered water (1.88 mg L$^{-1}$), and UV$_{254}$ absorbance was 0.17 cm$^{-1}$ at B, which was 7.7 times higher on average than 0.022 cm$^{-1}$ of N. The color of B-filtered water was high at 15°; in contrast, alkalinity and T-N were lower than in N-filtered water. The •OH scavenging demand was 20,659 s$^{-1}$ in N-filtered water and 82,044 s$^{-1}$ in B-filtered water. These results of the filtered water quality of the N and B water treatments are shown in Table 3.

**Table 3.** Characteristics of target water quality in this study.

| Parameter | N-Filtered Water | B-Filtered Water |
|---|---|---|
| pH | 7.2 ± 0.2 | 6.9 ± 0.3 |
| Alkalinity (mg L$^{-1}$ as CaCO$_3$) | 43 ± 4 | 30 ± 2 |
| Turbidity (NTU) | 0.13 ± 0.10 | 0.31 ± 0.20 |
| TOC (mg L$^{-1}$) | 1.88 ± 0.32 | 6.96 ± 0.53 |
| TDS (mg L$^{-1}$) | 111 ± 3 | 133 ± 3 |
| UV254 (cm$^{-1}$) | 0.022 ± 0.003 | 0.170 |
| Color (°) | 0 ± 1 | 15 ± 3 |
| T-N (mg L$^{-1}$) | 2.2 ± 0.3 | 1.3 ± 0.5 |
| •OH scavenging demand (s$^{-1}$) | 20,659 ± 4907 | 82,044 ± 5071 |

The •OH radical demand was continuously monitored for one year, and it was confirmed that it changed according to seasonal variations and water characteristics. In a previous study, Kwon monitored seasonal variability of •OH water demand data in Han-River [31]. The main substances known as •OH scavengers in water are organic substances, carbonates, bicarbonates, nitrates, and inorganic species such as bromide ions [27]. Previous studies have reported that up to 95% of the •OH scavengers originated from organic matter [18].

*3.2. Comparison of Degradation Rate Constants for Reactions of PhACs with UV/$H_2O_2$ under Different Conditions of •OH Scavenging Demand*

Among the 16 target PhACs, atenolol, caffeine, carbamazepine, and sulfamethoxazole were selected to compare the effect of the scavenging demand on the removal rate in N and B water treatment plants. These were purchased from CHIRON (1 mL, 1000 µg mL$^{-1}$ in methanol, Norway). Table 4 lists the degradation rate constants of the PhACs in N and B. The decomposition rate of each target compound represents the slope obtained as a function of UV irradiation and the decomposition rate of the target compounds. The decomposition rates were compared by varying the concentration of $H_2O_2$ from 0 to 10 mg L$^{-1}$. The decreasing trend of the target PhAC concentrations with increasing UV irradiation can be interpreted by the pseudo-first-order reaction (Equation (2)) [32].

$$-\frac{d[\text{PhAC}]}{dt} = (kd + ki) \times [\text{PhAC}] = \{kd + (k_{\bullet\text{OH,PhAC}}[\bullet\text{OH}])\} \times [\text{PhAC}] \quad (2)$$

where *kd* is the time-based pseudo-first-order rate constant of targeting PhAC degradation by direct UV photolysis (s$^{-1}$), *ki* is the time-based pseudo-first-order rate constant of targeting PhAC degradation by •OH reactions (s$^{-1}$), $k_{\bullet\text{OH,PhAC}}$ is the second-order rate constant between •OH and targeting PhACs.

Integrating Equation (2) and dividing by the UV intensity (mW cm$^{-2}$) and time (*t*) yields Equation (3):

$$\frac{\ln([\text{PhAC}]_0/[\text{PhAC}]_t)}{E_0 \cdot t} = \text{kd} + \text{ki} = \text{kd} + \frac{k_{\bullet\text{OH,PhAC}} \int_0^t [\bullet\text{OH}]dt}{E_0 \cdot t} = \text{kT} \quad (3)$$

where kT is the fluence-based pseudo-first-order rate constant of PhAC degradation by both direct UV photolysis and •OH reaction.

The UV irradiation dosage used in this experiment was 1980 mJ cm$^{-2}$, and the injection amount range of $H_2O_2$ for the analysis was 0–10 mg L$^{-1}$. The pH was adjusted to 7, and the UV intensity was 1.1 mW cm$^{-2}$. The concentration of PhACs was 200 ng L$^{-1}$. While the degradation rate constant for atenolol was $6.27 \times 10^{-4}$ cm$^2$ mJ$^{-1}$, and $5.03 \times 10^{-4}$ cm$^2$ mJ$^{-1}$ for the N and B target source water, respectively, the degradation rate constant for caffeine was found to be $5.11 \times 10^{-4}$ cm$^2$ mJ$^{-1}$, and $3.95 \times 10^{-4}$ cm$^2$ mJ$^{-1}$, respectively. Carbamazepine showed a degradation rate constant of $7.60 \times 10^{-4}$ cm$^2$ mJ$^{-1}$, and $5.58 \times 10^{-4}$ cm$^2$ mJ$^{-1}$ for each source water. The degradation rate constants of sulfamethoxazole were $27.98 \times 10^{-4}$ cm$^2$ mJ$^{-1}$, and $22.66 \times 10^{-4}$ cm$^2$ mJ$^{-1}$, respectively. The removal rate constants of all four PhACs were lower in B plants, which could be attributed to the elimination of •OH by NOM in water [33]. In the case of B-filtered water, the •OH scavenging demand was 4.2 times higher than that of N-filtered water because the concentration of organic compounds and chromaticity in B plants were relatively high. The presence of high concentration organic compounds reduces the concentration of •OH that can react with target compounds [34]. This means that the •OH scavenging demand of water is an essential factor influencing the removal rate constants, and since scavenging substances exist differently from water source to source, quantitative interpretation of •OH scavenging demand is significant.

**Table 4.** PhACs degradation rate constants: compared results for each of the substances. $[PhACs]_0 = 200$ ng $L^{-1}$, $[H_2O_2]_0 = 0, 2, 5, 10$ mg $L^{-1}$, [UV dose] = 1980 mJ cm$^{-2}$.

| | H$_2$O$_2$ (mg L$^{-1}$) | Degradation Rate Constants (k, $\times 10^{-4}$, cm mJ$^{-1}$) | | | |
| --- | --- | --- | --- | --- | --- |
| | | Atenolol | Caffeine | Carbamazepine | Sulfamethoxazole |
| N-filtered water | 0 | 0.09 | 0.82 | 0.35 | 22.23 |
| | 2 | 1.52 | 0.89 | 1.84 | 23.02 |
| | 5 | 3.18 | 2.55 | 3.87 | 24.22 |
| | 10 | 6.27 | 5.11 | 7.60 | 27.98 |
| B-filtered water | 0 | 0.24 | 0.12 | 0.30 | 19.31 |
| | 2 | 1.14 | 0.72 | 1.44 | 19.50 |
| | 5 | 2.53 | 2.16 | 2.91 | 20.31 |
| | 10 | 5.03 | 3.95 | 5.58 | 22.66 |

*3.3. Categorization of Groups of PhACs According to Their Decomposition Properties*

In this study, the grouping of target PhACs was proposed. The selected PhACs were classified into three groups depending on their relative reactivity to UV direct photolysis and •OH oxidation. They were divided according to their vulnerability to direct photolysis. When selecting groups, oxidants and UV doses calculated in the nonlinear model were considered in Equation (4) [31]. The target removal rate of the PhACs was set at 90%.

$$\ln\left(\frac{[M]_0}{[M]_{H'}}\right) = H' \times (k_d' + k_i') = H' \times \left\{ \frac{\varepsilon_M \cdot \phi_M \cdot \ln(10)}{U_{254}} + k_{M,\bullet OH} \frac{\varepsilon_{H2O2} \cdot \Phi_{\bullet OH} \cdot [H_2O_2] \cdot \ln(10)}{U_{254} \cdot (\sum k_{s,OH}[S]_i + k_{H2O2,OH}[H_2O_2])} \right\} \quad (4)$$

where $H'$ is the UV fluence (mJ cm$^{-2}$), $[M]$ is the molar concentration of the target model compound (mol L$^{-1}$), $k_{M,\bullet OH}$ is the second-order rate constant for the reaction of •OH with the target compound M, $k_d'$ is the pseudo-first-order rate constant for direct UV photolysis (cm$^2$ mJ$^{-1}$), $k_i'$ is the pseudo-first-order rate constant for •OH induced degradation (cm$^2$ mJ$^{-1}$), $\varepsilon$ is the molar absorption coefficient (M$^{-1}$ cm$^{-1}$), $\Phi_{\bullet OH}$ is the quantum yield of •OH production from H$_2$O$_2$ photolysis, $[H_2O_2]$ is the concentration of H$_2$O$_2$ (M), and $\sum k_{s,OH}[S]_i$ is the •OH scavenging demand (s$^{-1}$).

The •OH scavenging demand used in the modeling condition for the grouping was 82,044 s$^{-1}$. Group 1 includes photo-susceptible PhACs, easily photodegraded with no additional oxidants or minor degradation by •OH. Group 2 consisted of moderate photodegradable compounds with high reactivity for •OH oxidation. Group 3 consisted of photo-resistant PhACs. The grouping of target compounds was presented in a previous study [24]. The PhACs corresponding to each group is shown in Table 5.

**Table 5.** PhACs classification by removal characteristics.

| PhACs Classification by Removal Characteristics | | |
| --- | --- | --- |
| Group 1 | Group 2 | Group 3 |
| Acetaminophen | Bisphenol A | |
| Amoxicillin | Carbamazepine | Atenolol |
| Diclofenac | Ibuprofen | Atrazine |
| Iopromide | Naproxen | Caffeine |
| Ketoprofen | Ciprofloxacin | Nitrobenzene |
| Sulfamethoxazole | Tetracycline | |

In Group 1 (acetaminophen, amoxicillin, diclofenac, iopromide, ketoprofen, and sulfamethoxazole), a target removal rate of 90% was achieved even when irradiated with UV alone [13]. Photodegradation efficiency depends on parameters such as the molar absorption coefficient ($\varepsilon$) and quantum yield ($\Phi$) at the wavelength ($\lambda$) [35]. Because the

photon energy of the UV irradiation applied to the water is assumed to be almost constant, the decomposition rate of the compound by UV direct photolysis is proportional to the photodegradation of the target compound, which can be defined by multiplying by $\varepsilon$ and $\Phi$. PhACs in group 1 have higher $\varepsilon$ and $\Phi$ values than other groups, so they showed higher photo-degradability values [24,36].

The degradation efficiencies of group 2 (bisphenol A, carbamazepine, ibuprofen, naproxen, ciprofloxacin, and tetracycline) were dominated by both direct UV photolysis and •OH oxidation. In contrast to group 1, direct photolysis was insignificant for the PhACs in group 3 (atenolol, atrazine, caffeine, and nitrobenzene). This could be expected from their low photo degradation (low molar absorption coefficients and quantum yields (Table 1).

### 3.4. Optimization of Operating Conditions for PhACs Degradation

In almost all cases, the selection of the optimum operating conditions for $UV/H_2O_2$ is usually driven by energy consumption. The energy consumption can differ depending on the combination of UV dose and $H_2O_2$ concentration. Therefore, it is crucial to investigate the optimal operating parameters to achieve each targeting removal rate while minimizing energy consumption. Bolton et al. proposed the electrical energy per order (EEO, unit: kWh m$^{-3}$ order$^{-1}$) concept that was used in evaluating the electrical energy efficiency to treat the target compound [37]. The electrical energy demand (EED) is derived from the EEO concept as shown in Equations (5) and (6).

$$\text{EED} \left( \frac{\text{kWh}}{\text{m}^3} \right) = \frac{(P \times T)}{60V} \tag{5}$$

$$\text{EEO} \left( \frac{\text{kWh}}{\text{m}^3} \right) = \frac{EED}{\left( \log \left( \frac{M_0}{M_e} \right) \right)} \tag{6}$$

where $P$ is the power (kW), $T$ is the irradiation time (min), $V$ is the total system volume (m$^3$), and $M_0$ and $M_e$ are the concentrations before and after the UV reactor, respectively.

The target EED (kWh m$^{-3}$) was determined by multiplying the EEO value by the target Log10 reduction (order). EED is also calculated by the sum of the electrical energy cost incurred by the UV treatment and oxidant-related costs [14,31,36,38]. In this study, the EED values (kWh m$^{-3}$) were calculated to estimate the energy consumption of $UV/H_2O_2$, as shown in Equation (7).

$$\text{EED}_{UV/H2O2} = \text{EED}_{UV} + \text{EED}_{H2O2} = H' \times \left( \frac{2.75 \cdot 10^{-7} \cdot \left( 10 \cdot H'/WF \right)}{l^{avg} \cdot \eta_{UV}} \right) + \left( a \times [H_2O_2]_0 \right) \tag{7}$$

where $\text{EED}_{UV}$ is the electrical energy dose associated with the UV reactor power requirements and $\text{EED}_{H2O2}$ is the electrical energy dose required for the production of $H_2O_2$ dosed in the $UV/H_2O_2$ process (kWh m$^{-3}$), 1 Joule is $2.78 \times 10^{-7}$ kWh, $l^{avg}$ is the average optical path length of the reactor, $\eta_{UV}$ is the UV lamp efficiency and $[H_2O_2]_0$ is the $H_2O_2$ dose (kg m$^{-3}$), a is the energy requirements to produce 1 kg $H_2O_2$. WF is the water factor, which accounts for the effect of $a_{254nm}$ on $\text{EED}_{UV}$.

The removal rate of the target PhACs was affected by the •OH scavenging demand of the target source water. Therefore, it is necessary to optimize the operating parameters of $UV/H_2O_2$ under the consideration of the •OH scavenging demand of the target source water. The average •OH scavenging demand of the N- and B-filtered water used in the calculations was 20,659 s$^{-1}$ and 82,044 s$^{-1}$.

Figure 2 shows the EED contours of selected representative substances belonging to groups 1, 2, and 3 (sulfamethoxazole, carbamazepine, and nitrobenzene, respectively). As shown in Figure 2, the removal curves of target substances by $UV/H_2O_2$ showed a nonlinear shape; thus, the generalized reduced gradient nonlinear solver method was applied in this study. Using this method, we calculated the conditions for UV and chemical

injection to simultaneously minimize the energy consumption and achieve the target removal rate for each target substance under the conditions of •OH scavenging demand of the target source water. The range of UV dose for the analysis was 0 to 1200 mJ cm$^{-2}$ for the N treatment and 0 to 2000 mJ cm$^{-2}$ for the B treatment. The oxidant injection dosage used in the analysis was 0–10 mg L$^{-1}$. The concentration of the target compound was set at 100 ng L$^{-1}$, and the target removal rate of the target compound was set at 90%. The optimal operating conditions to achieve 90% removal of the target compound in the N and B treatments are presented in Table 6. For group 1, the EED values in N-filtered water and B-filtered water, optimal UV, and H$_2$O$_2$ injection amounts were similar. This is because the PhACs of group 1, can be easily photo-decomposed without decomposition by •OH.

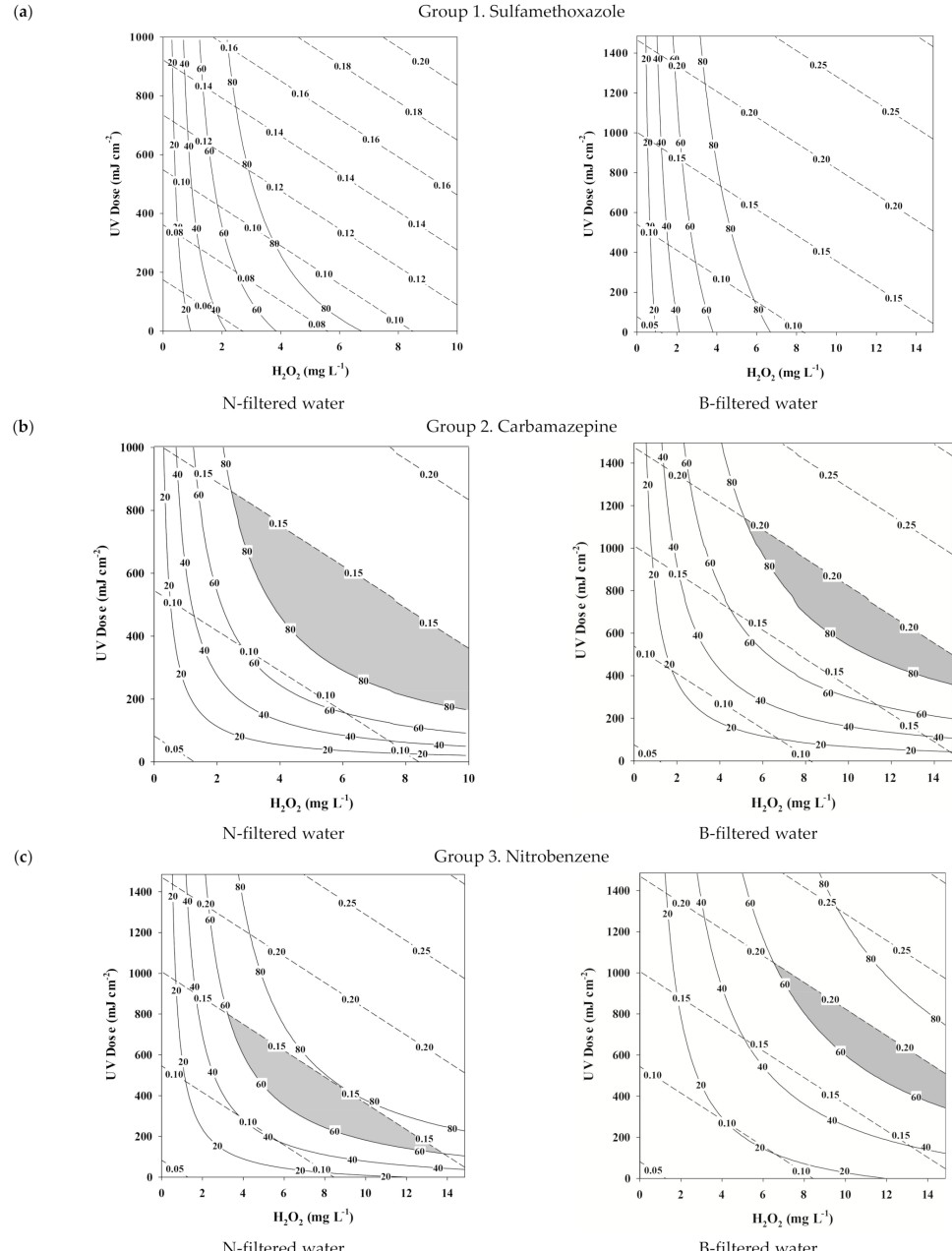

**Figure 2.** Contours of the EED value and removal efficiency at various UV doses and H$_2$O$_2$ concentrations: (**a**) group 1; sulfamethoxazole, (**b**) group 2; carbamazepine, (**c**) group 3; nitrobenzene.

**Table 6.** EED and optimal injection dosage of $H_2O_2$ and UV for 90% removal of each PhACs in the UV/$H_2O_2$.

| | | N Plant | | | B Plant | | |
|---|---|---|---|---|---|---|---|
| | UV254 | | 0.0210 | | | 0.0124 | |
| | •OH scavenging demand ($s^{-1}$) | | 20,659 | | | 82,044 | |
| | pH | | 7.2 | | | 7.7 | |
| | | UV dose for 90% Removal (mJ $cm^{-2}$) | $H_2O_2$ for 90% Removal (mg $L^{-1}$) | EED for 90% Removal (kWh $m^{-3}$) | UV dose for 90% Removal (mJ $cm^{-2}$) | $H_2O_2$ for 90% Removal (mg $L^{-1}$) | EED for 90% Removal (kWh $m^{-3}$) |
| Group 1 | Acetaminophen | 237 | 0 | 0.058 | 211 | 0 | 0.056 |
| | Amoxicillin | 757 | 1.39 | 0.109 | 1072 | 0 | 0.116 |
| | Diclofenac | 347 | 0 | 0.065 | 343 | 0 | 0.065 |
| | Iopromide | 590 | 0 | 0.082 | 584 | 0 | 0.0820 |
| | Ketoprofen | 666 | 0 | 0.088 | 105 | 0 | 0.049 |
| | Sulfamethoxazole | 744 | 1.06 | 0.105 | 957 | 0 | 0.1080 |
| Group 2 | Bisphenol A | 678 | 3.80 | 0.130 | 1247 | 7.43 | 0.208 |
| | Carbamazepine | 510 | 2.96 | 0.109 | 1246 | 7.44 | 0.209 |
| | Ibuprofen | 702 | 3.70 | 0.130 | 1290 | 6.78 | 0.205 |
| | Naproxen | 639 | 3.15 | 0.120 | 1179 | 5.26 | 0.180 |
| | Ciprofloxacin | 766 | 3.02 | 0.127 | 1397 | 3.56 | 0.171 |
| | Tetracycline | 689 | 3.72 | 0.130 | 1266 | 6.99 | 0.205 |
| Group 3 | Atenolol | 721 | 3.91 | 0.134 | 1325 | 7.39 | 0.214 |
| | Atrazine | 1276 | 4.09 | 0.174 | 1500 | 10 | 0.254 |
| | Caffeine | 769 | 4.23 | 0.141 | 1405 | 8.23 | 0.229 |
| | Nitrobenzene | 1094 | 5.36 | 0.175 | 1954 | 9.70 | 0.282 |

In the case of groups 2 and 3, it was confirmed that they had a higher EED value in B-filtered water with much organic matter, and the optimal UV dose and $H_2O_2$ concentration were also high. In the case of carbamazepine, the representative indicator of group 2, the optimized operating conditions were UV 510 mJ $cm^{-2}$ and $H_2O_2$ 2.96 mg $L^{-1}$ for the N plant but UV 1246 mJ $cm^{-2}$ and $H_2O_2$ 7.44 mg $L^{-1}$ for the B plant. The EED value was also 0.1088 kWh $m^{-3}$ for the N plant and 0.2085 kWh $m^{-3}$ for the B plant, showing two times higher values compared with the N plant. In the case of nitrobenzene, the representative indicator of group 3, the optimized operating conditions were UV 1094 mJ $cm^{-2}$ and $H_2O_2$ 5.36 mg $L^{-1}$ for the N plant, UV 1,954 mJ $cm^{-2}$ and $H_2O_2$ 9.70 mg $L^{-1}$ for the B plant. The EED value was also 0.1754 kWh $m^{-3}$ in the N plant and 0.2823 kWh $m^{-3}$ in the B plant.

Figure 2 shows the contours of specific energy and removal efficiency for the target compound in each group as a function of UV dose and $H_2O_2$ concentration. Here, the two contours were overlaid to determine the optimum UV dose and $H_2O_2$ concentration for a given setting value. In the case of group 2, if the energy consumption and removal rates have to be less than 0.15 kWh $m^{-3}$ and higher than 80%, respectively, the UV dose and $H_2O_2$ concentration should range from 200 to 840 mJ $cm^{-2}$ and 2.5 mg $L^{-1}$ or more for the N plant. The UV dose and $H_2O_2$ concentration should range from 400 to 1170 mJ $cm^{-2}$ and 5.3 mg $L^{-1}$ or more for the B plant. In the case of group 3, if the energy consumption and removal rates have to be less than 0.20 kWh $m^{-3}$ and higher than 60%, respectively, the UV dose and $H_2O_2$ concentration should range from 150 to 800 mJ $cm^{-2}$ and from 3.0 mg $L^{-1}$ to 13.8 mg $L^{-1}$ for the N plant. The UV dose and $H_2O_2$ concentration should

range from 350 to 1070 mJ cm$^{-2}$ and 6.2 mg L$^{-1}$ or more for the B plant. In group 3, if the target removal rate is high, excess H$_2$O$_2$ should be injected and excess energy needed. In both groups, compared with the same target removal rate, the optimized operating conditions of the B plant were higher than the N plant. In that case, excess H$_2$O$_2$ reacts with •OH to generate hydro-peroxyl radical (1.70 eV) with low oxidation power, hence reducing removal efficiency [16]. In addition, if the concentration of hydrogen peroxide remaining in the treated water is high, the cost of the post-treatment should be increased. For this, the use of non-oxidizing agents and an activated carbon process must be installed as a post-treatment. Considering seasonal variations for •OH scavenging demand in raw water sources, it is evident from these results that •OH scavenging demand is an important factor affecting the optimization of the UV/H$_2$O$_2$ process. Therefore, when operating the UV/H$_2$O$_2$ process, the operating conditions must be determined by the •OH scavenging demand and the targeting energy consumption.

## 4. Discussion

The types and concentrations of inorganic species such as DOM, carbonates, bicarbonates, nitrates, and bromide ions are critical water quality parameters that influence operation parameters of the UV/H$_2$O$_2$ oxidation process. In addition, there are cases of detection reports of targeted PhACs in water sources for water purification plants in Korea, and trace organic compounds include iopromide, ibuprofen, caffeine, and naproxen. This study proposed the model predictive control method to remove the target compound considering energy consumption and •OH scavenging demands. The previous study proposed a method of continuously monitoring the •OH scavenging demands, and real-time model predictive control is possible if the monitoring value is linked to the model. In this study, the target compounds were divided into three groups, and the optimal UV dose and H$_2$O$_2$ concentration range were presented for each group. In the previous study [20], caffeine was included in group 2 (group 1 (k$_{•OH}$/k$_{UV}$ < 0.1); group 2 (0.1 ≤ k$_{•OH}$/k$_{UV}$ < 1); and group 3 (k$_{•OH}$/k$_{UV}$ ≥ 1)), but in this study, it was classified as group 3. The difference with the previous paper is that this study was grouped based on the optimal UV dose and H$_2$O$_2$ concentration derived by the analyzed •OH scavenging demands.

## 5. Conclusions

In this study, •OH scavenging demand analyzer was monitored year-round at two water purification plants, N and B. This study proposed three groups based on the simulated UV dose and H$_2$O$_2$ concentration considering the •OH scavenging demand. Using our model, the optimum UV dose and H$_2$O$_2$ concentration can be determined for a given condition of targeting energy consumption and removal rates. The B plant generally showed higher EED values and required higher UV and H$_2$O$_2$ injection amounts than the N plant. The EED values obtained as a result of modeling were as follows: For group 2, N plant: 1088–1302 kWh m$^{-3}$; B plant: 1712–2085 kWh m$^{-3}$. Based on the overlaid contours analysis for group 2, the optimal range of UV dose and H$_2$O$_2$ concentration were from 200 to 840 mJ cm$^{-2}$ and 2.5 mg L$^{-1}$ or more for the N plant. However, the optimal range of UV dose and H$_2$O$_2$ concentration were from 400 to 1170 mJ cm$^{-2}$ and 5.3 mg L$^{-1}$ or more for the B plant. It was found that the •OH scavenging demand is an important factor to determine the optimization of the UV/H$_2$O$_2$ process.

**Author Contributions:** Conceptualization, T.-M.H.; methodology, J.L. and T.-M.H.; investigation, J.L.; validation, T.-M.H. and J.-W.K.; formal analysis, S.-H.N.; resources, T.-M.H.; data curation, T.-M.H. and E.K.; writing—original draft preparation J.L.; writing—review and editing, T.-M.H.; visualization, E.K.; supervision, T.-M.H.; project administration, T.-M.H.; funding acquisition, T.-M.H. All authors have read and agreed to the published version of the manuscript.

**Funding:** This work was supported by the Korea Environment Industry & Technology Institute (KEITI) through an environmental R&D project for developing innovative drinking water and wastewater technologies program, funded by the Korea Ministry of Environment (MOE) (grant number 2020002690003 and 2020002700004).

**Institutional Review Board Statement:** Not applicable.

**Informed Consent Statement:** Not applicable.

**Data Availability Statement:** Not applicable.

**Acknowledgments:** Not applicable.

**Conflicts of Interest:** The authors declare no conflict of interest.

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
