# Peer review of "Model Predictive Control Strategy for the Degradation of Pharmaceutically Active Compounds by UV/H2O2 Oxidation Process"

_water, doi:10.3390/w14030385_

Round 1

Reviewer 1 Report

This manuscript contains many English errors (including the title). This reviewer will not correct them. If the manuscript moves toward publication, the authors should be required to find someone fluent in scientific English to correct the manuscript. This reviewer has corrected the abstract as a guide to the authors

Abstract: Hydroxyl radical (•OH) scavenging demand can be an indicator that represents the water quality characteristics of the raw water. It is one of the key parameters predicting UV/ H2O2 system performance and affects the operating parameters. Based on the •OH scavenging demand, we have developed the a model predictive control strategy to meet the target compound removal efficiency and energy consumption simultaneously. The Selected pharmaceutically active compounds (PhACs) were classified into three groups depending on the reaction rates UV direct photolysis and of UV and susceptibility to •OH radical attack arising from H2O2 photolysis. Group 1 for photo-susceptible PhACs; Group 2 for PhACs susceptible to both direct photolysis and •OH oxidation; and Group 3 for photo-resistant PhACs. In the case of group 2, the optimized operating parameters ranges of UV and H2O2 were as follow (N plant: UV dose 510-702 mJ cm-2, H2O2 dose 2.96-3,80 mg L-1; B plant: UV dose 1,179-1,397 mJ cm-2, H2O2 dose 3.56-7.44 mg L-1). In the case of group 3, the operating parameters ranges were as follow (N plant: UV dose 721-1,276 mJ cm-2, H2O2 dose 3.02-5.36 mg L-1; B plant: UV dose 1,325-1,954 mJ cm-2, H2O2 dose 7.39-10.0 mg L-1). The model was also applied to operating control strategy for low energy requirements and target compound removal.

{the parameter ranges should not be expressed with so many significant figures.}

Line 50 and following – the authors have not cited any reviews of UV/H2O2 AOP. In particular, they have not cited any of the excellent reviews in M. Stefan, Ed., Advanced Oxidation Processes for Water Treatment: Fundamentals and Applications, 2017, IWA. Also, they have ignored the prior very important contributions by authors, such as Karl Linden, Michael Watts, James Bolton, J. C. Crittenden, etc.

Line 107 and following – there is no description of the UV reactor nor is there any information on how irradiances were determined

Lines 142, 143 – these are absorption coefficients

Line 145 – are these numbers really known to 5 significant digits?

Table 3 – last line – the authors clearly do not understand how ridiculous it is to quote errors to 5 significant digits!

Line 209 – the correct term is molar absorption coefficient

Section 3.4 – again the authors have ignored important prior citations. Here mention should have been made of the electrical energy per order (EEO) concept (see Bolton et al., Pure Appl. Chem. (2001) 73(4), 627-637.).

General Comments:

This reviewer is annoyed by poor literature citations, missing important information about the UV reactor and irradiance measurements, and very poor understanding of significant digits.

Author Response

- Review 1

This manuscript contains many English errors (including the title). This reviewer will not correct them. If the manuscript moves toward publication, the authors should be required to find someone fluent in scientific English to correct the manuscript. This reviewer has corrected the abstract as a guide to the authors

: Thank you for your reviewing the manuscript. According to your advice, we revised the title and manuscript. Our paper was once again edited by experts.

  1. Line 50 and following – the authors have not cited any reviews of UV/H2O2 AOP. In particular, they have not cited any of the excellent reviews in M. Stefan, Ed., Advanced Oxidation Processes for Water Treatment: Fundamentals and Applications, 2017, IWA. Also, they have ignored the prior very important contributions by authors, such as Karl Linden, Michael Watts, James Bolton, J.

Authors’ response: 

We revised as your comment. (Lines 51~57, 100~106) The papers you mentioned are also based on our study and were included in the cited paper, so they were not indicated separately. Thank you for pointing out the important parts.

  1. Line 107 and following – there is no description of the UV reactor nor is there any information on how irradiances were determined UV

Authors’ response: 

We added picture of continuous •OH scavenging demand analyzer in Figure 1. The following content was added below the 109 line. “The incident irradiance measurements were obtained by placing a calibrated radiometer (UVX Radiometer, UVP Co., USA) at the height of the water level in the Petri dish to obtain the average incident irradiance across the solution surface using the petri dish factor (PF) and reflection factor (RF) [30].” (Lines 112~119)

  1. Lines 142, 143 – these are absorption coefficients

Authors’ response:

The UV254 absorbance was 0.17 cm-1 at B plant, which was 7.7 times higher on average than 0.022 cm-1 of N plant. (Line147)

  1. Line 145 – are these numbers really known to 5 significant digits?

Authors’ response:

Details of •OH scavenging demand can be found in my previous paper. (Kwon et al, Environ. Sci. Technol. 2019, 53, 3177–3186) [31]. It is known that there is a large variation of •OH water demand across the water sources. (Rosenfeldt et al, Environ. Sci. Technol. 2007, 41 (7), 2548−2553, Kwon et al, Chemical Engineering Journal 2014, 236, 438–447) There are limited literature study focused and reported on the seasonal variation of the •OH water demand. In Kwon’s paper, the measured •OH water demand mostly fell within the 1.6 × 104 − 2.7 × 104 s−1 range. In this study, we compared the •OH water demand variation at two water sources. The •OH water demand was monitored for year-round and demonstrate its dependency upon seasonal variation. We evaluated and compared the values with the Kwon’s paper. (Kwon et al, Environ. Sci. Technol. 2019, 53, 3177–3186)

  1. Line 145 – Table 3 – last line – the authors clearly do not understand how ridiculous it is to quote errors to 5 significant digits!

Authors’ response:

We checked the original calculation sheet again. A typo error was found in the data sheet used for the standard deviation calculation. We revised the value for plant B in Table 3.

  1. Line 209 – the correct term is molar absorption coefficient

Authors’ response:

We revised it as your comment. We checked the term.

  1. Section 3.4 – again the authors have ignored important prior citations. Here mention should have been made of the electrical energy per order (EEO) concept (see Bolton et al., Pure Appl. Chem. (2001) 73(4), 627-637

Authors’ response:

The paper you mentioned is also based on our study. Accordingly, the description for EEO or EED model was added in manuscript and reference part [30, 31]. (Lines 245~250)  

General Comments:

This reviewer is annoyed by poor literature citations, missing important information about the UV reactor and irradiance measurements, and very poor understanding of significant digits.

Authors’ response:

Important references were cited by referring to the opinions of the review, additional explanations were given for the mentioned contents, and the reasons for the digits were explained. Thank you again.

Reviewer 2 Report

This manuscript demonstrates a prediction model for UV/H2O2 process in the mitigation of pharmaceutically active compounds. Although there are many typo errors, the manuscript provides a new perspective and appoarch to predict OH scavenging demand, which is particularly needed in the engineering water treatment. Therefore, I would like to recommend a major revision before its publication. Specific comments are shown as following:

  1. Line 13, additional blank, a typo error, occurred in "UV/ H2O2".
  2. In the abstract, the authors should briefly summarize the specific compound of Group 1, 2, and 3, for information, there are too many of photo-suspectible PhACs to be focused.
  3. Line 46-49, the examples of "advanced drinking water treatment processes" is untypical. The ozonation and activated carbon adsorption are very conventional and commly used in water treatment. However, advanced water treatment processes such as catalytic ozonation (https://doi.org/10.1007/s12274-021-3918-6; Chemical Engineering Journal, 2021, 404, 127075) and Fenton oxidation (Applied Catalysis B: Environmental, 2021, 286, 119859) should be added at least.
  4. Line 79, delete "in this study".
  5. Point data should be presented along with the table 3.

Author Response

  1. Line 13, additional blank, a typo error, occurred in "UV/ H2O2".

Authors’ response:

 We revised it as your comment.

  1. In the abstract, the authors should briefly summarize the specific compound of Group 1, 2, and 3, for information, there are too many of photo-suspectible PhACs to be focused.

Authors’ response:

According to your comment, the substances for each groups were added to the abstract. (Lines 21~24)

  1. Line 46-49, the examples of "advanced drinking water treatment processes" is untypical. The ozonation and activated carbon adsorption are very conventional and commly used in water treatment. However, advanced water treatment processes such as catalytic ozonation (https://doi.org/10.1007/s12274-021-3918-6; Chemical Engineering Journal, 2021, 404, 127075) and Fenton oxidation (Applied Catalysis B: Environmental, 2021, 286, 119859) should be added at least.

Authors’ response:

We revised manuscript and added the references as you recommended.

  1. Line 79, delete "in this study".

Authors’ response:

We revised it as your comment.

  1. Point data should be presented along with the table 3.

Authors’ response:

The mention of Table 3 has been revised to be presented along with the explanation of the data.

Reviewer 3 Report

In this manuscript, the OH scavenging demands of two water purification plants, N and B, were monitored using a hydroxyl radical scavenging demand analyzer. In this system, UV/H2O2 performance is predicted, and this has an impact on operating parameters. Overall, this manuscript is well organized and written. Interesting results were well presented. The length of the manuscript is appropriate. Discussion is detailed and gives answers to every aspect of numerous experiments applied in this work. I recommended accepting the paper to be published after language revisions and minor amendments:

  1. I recommend that the authors revise the abstract to provide more information on the background, objectives, methods, main findings, and conclusion. Please include a phrase demonstrating the importance of the research.
  2. Conclusions – Need to rewrite. The current version is too long. The key results are not highlighted.
  3. Much more explanations and interpretations must be added for the results, which are not enough.
  4. The chemicals and their purity should be provided in the experimental section

Author Response

  1. I recommend that the authors revise the abstract to provide more information on the background, objectives, methods, main findings, and conclusion. Please include a phrase demonstrating the importance of the research.

Authors’ response:

Thank you for your reviewing the manuscript

According to your advice, I revised the abstract. (Lines18-26) We added the main findings as follows. ‘It was confirmed that the optimal operating conditions and EED values changed according to the •OH scavenging demand.’

  1. Conclusions – Need to rewrite. The current version is too long. The key results are not highlighted.

Authors’ response:

We revised the conclusion as your comment. (Lines 352~360).

  1. Much more explanations and interpretations must be added for the results, which are not enough.

Authors’ response:

Section 3.1

We added the results and references for more explanation of •OH scavenging. (Line 147~152 and Line 238~257) The •OH radical demand was continuously monitored for around one year, and it was confirmed that it changed according to seasonal variations and water characteristics.

Section 3.3

: We changed the model equation for more clarity. And we revised the explanation of categorization for PhACs. (Lines 205~228)

Section3.4

: The description for EEO or EED model was added in manuscript and reference part [30, 31]. (Lines 242~247)  More explanations were reflected of the overall manuscript.

  1. The chemicals and their purity should be provided in the experimental section

Authors’ response:

We added the material information as your comment (Line 90, 162~163)

Material information was added to section 2.1 and 3.2.

Round 2

Reviewer 1 Report

The units for EEO are kWh/order/m3

Reviewer 2 Report

This manuscript has been improved. I have no more comments.